# Hypertension Associated with Fructose and High Salt: Renal and Sympathetic Mechanisms

**DOI:** 10.3390/nu11030569

**Published:** 2019-03-07

**Authors:** Dragana Komnenov, Peter E. Levanovich, Noreen F. Rossi

**Affiliations:** 1Department of Physiology, Wayne State University, 4160 John R Street #908, Detroit, MI 48201, USA; dkomneno@med.wayne.edu (D.K.); plevanov@med.wayne.edu (P.E.L.); 2Department of Internal Medicine, Wayne State University, 4160 John R Street #908, Detroit, MI 48201, USA; 3John D. Dingell VA Medical Center, 4646 John R Street, Detroit, MI 48201, USA

**Keywords:** fructose, hypertension, renin-angiotensin-aldosterone system, renal transporters, sodium, renal sympathetic nerve activity

## Abstract

Hypertension is a leading cause of cardiovascular and chronic renal disease. Despite multiple important strides that have been made in our understanding of the etiology of hypertension, the mechanisms remain complex due to multiple factors, including the environment, heredity and diet. This review focuses on dietary contributions, providing evidence for the involvement of elevated fructose and salt consumption that parallels the increased incidence of hypertension worldwide. High fructose loads potentiate salt reabsorption by the kidney, leading to elevation in blood pressure. Several transporters, such as NHE3 and PAT1 are modulated in this milieu and play a crucial role in salt-sensitivity. High fructose ingestion also modulates the renin-angiotensin-aldosterone system. Recent attention has been shifted towards the contribution of the sympathetic nervous system, as clinical trials demonstrated significant reductions in blood pressure following renal sympathetic nerve ablation. New preclinical data demonstrates the activation of the renal sympathetic nerves in fructose-induced salt-sensitive hypertension, and reductions of blood pressure after renal nerve ablation. This review further demonstrates the interplay between sodium handling by the kidney, the renin-angiotensin-aldosterone system, and activation of the renal sympathetic nerves as important mechanisms in fructose and salt-induced hypertension.

## 1. Introduction

Hypertension is a multifactorial condition rather than a single disease entity whose onset can be brought about through a variety of factors originating from environmental, dietary, or hereditary factors. The complex interplay of these components has frustrated our understanding of hypertension, despite decades of research spent attempting to identify its causes and develop viable treatments. Even today, less than 20% of all hypertensive cases have a known etiology (referred to as secondary hypertension) with the basis for the remaining majority of cases being unknown (referred to as primary or essential hypertension) [1,2]. Studies of the U.S. population have found that 29% of adults are hypertensive, and this number is only expected to increase dramatically in the near future [3]. Unfortunately, elevated blood pressure is becoming increasingly prevalent in people under the age of 40 years [4]. Despite the fact that the proportion of individuals with controlled blood pressure (systolic < 140 mmHg; diastolic < 90 mmHg) in the U.S. has increased from 28.4% to 43.5%, in low and middle-income countries it has actually decreased to 7.7% [5,6]. Indeed, even mild increases in either systolic or diastolic pressure (<10 mmHg) are accompanied by increases in mortality rates [7]. The increased prevalence of hypertension has coincided with an equally sharp increase in the incidence of chronic kidney disease that has been attributed, at least in part, to substantial changes in dietary intake and sedentary lifestyle [4,8]. Hence, there has been an increased effort to develop new models of hypertension that better reflect the environmental and dietary behaviors of modern society. Several models of diet-induced hypertension exist that are well established, namely those high in fat or sodium [9,10]. Models of metabolic syndrome induce a constellation of medical conditions accompanying insulin resistance such as obesity, hyperglycemia, hypertension and dyslipidemia [11]. Given the widespread use of fructose as a sweetener in food products, several recent reviews have discussed the impact of fructose ingestion on obesity and hypertension within the metabolic syndrome [12,13] and the hormones involved [14]. The present review focuses on recent interest in the consequences of even mild fructose consumption independent of full-blown metabolic syndrome with particular attention to the role of the kidney and the sympathetic nervous system on blood pressure.

Study selection: We searched PubMed (https://www.ncbi.nlm.nih.gov/pubmed/), the Cochrane Registry (http://www.cochranelibrary.com/about/central-landing-page.html), and the Web of Science Core Collection (https://www.library.ethz.ch/en/Resources/Databases/Web-of-Science-Core-Collection) from January 1975 to January 2019. The following search terms were used: fructose and blood pressure or hypertension; fructose and sodium; fructose and kidney; fructose and sympathetic nervous system. Then the search terms were refined with the addition of sodium to the search. Human and animal studies were included. This resulted in 428 articles whose abstracts were screened for relevancy to our topic of renal and sympathetic mechanisms involved in blood pressure control in the presence of fructose and high salt diets.

## 2. Fructose Consumption, Hypertension and Mortality

Total fructose consumption includes that which is found in high fructose corn syrup (HFCS) and sucrose. Generation of HFCS began in the late 1950s when it was discovered that glucose, hydrolyzed from corn starch extracts, could be partially converted to fructose through enzymatic isomerization [15]. Over time, this process was industrialized and led to the development of the corn-derived sweetener, HFCS, which can be synthesized with varying ratios of fructose to glucose content. Ease of synthesis, comparable flavor, and low cost of this ingredient have contributed to its widespread use as a sweetener [16]. Between 1970 and 2006, fructose consumption drastically increased, amounting to approximately 50% of all per capita added sugar consumption. This increase is accounted for solely by an exponential increase in free fructose consumption in the form of HFCS. During this time, caloric intake from total sugar (HFCS, sucrose, and other natural sugars) and fats increased significantly contributing to a 41% increase in total carbohydrate intake, of which fructose consumption provided a primary source [17].

HFCS appeals to nearly every demographic leading to its widespread use in food products. Statistical analysis of data collected in the National Health and Nutrition Examination Survey (NHANES I-III) determined that HFCS is most heavily consumed in the form soft drinks, and this trend is consistent throughout all age groups and sexes. Although a recent meta-analysis of three prospective studies fails to show incident hypertension associated with fructose, these studies relied on self-reports of physician-diagnosed hypertension and did not include concurrent sodium intake [18]. In contrast, another meta-analysis (*n* = 240,508) that includes data from the Coronary Artery Risk Development in Young Adults (CARDIA) cohort (*n* = 240,508) [19] and quoted by the recent American Heart Association update on stroke reports a 12% greater risk of hypertension with consumption of sugar sweetened beverages when controlled for sex, age, race, BMI and smoking behaviors [20]. Compared with the increased risk associated with more traditional factors such as alcohol consumption (61%), smoking (21%), and red meat intake (35%) and sedentary life style (48%) [21,22,23,24] or the non-traditional risk of stress (5–12%) [25], the risk associated with fructose intake may appear small but is nonetheless real. Whether the risk of hypertension associated with fructose is modified by combination with higher sodium intake has not yet been evaluated in humans, but the role of combined intake in preclinical studies is discussed below.

Adolescents and young adults are the highest consumers overall, and people in lower income population sectors are more likely to consume HFCS than those in more affluent demographic groups [13]. The rise in fructose consumption over the past several decades has been accompanied by an increase in obesity in the United States, and these rates parallel those of hypertension in a nearly linear relationship between body mass index and blood pressure [26,27]. Although several human studies have shown that high fructose consumption contributes to weight gain and blood pressure elevation, there is still controversy over the extent to which HFCS consumption is correlated with the historical obesity and hypertension trends [28]. Factors such as overall increase in national carbohydrate consumption make it challenging to discern increased fructose intake as a primary etiologic source for these disease states [17]. Nevertheless, the ingestion of fructose induces several physiologic responses that favor weight gain and increased blood pressure. 

The most recent NHANES III survey found that as of 2004, the average daily intake of fructose (49 g) in the U.S. equated to 9.1% of total energy intake [17]. Interestingly, commercially available soft drinks using HFCS have up to 140 calories from added sugars per 12 fluid ounce container. Given the most commonly used HFCS composition of 55% fructose and 45% glucose, this amounts to approximately 25 g or 100 calories from fructose alone. This quantity from one drink alone nearly surpasses the American Heart Association recommendation of only 150 and 100 calories from added sugars per day for men and women, respectively [29]. Animal studies designed to model this trend in human dietary intake have used various dietary fructose compositions—many of which exceed 60% of total daily caloric intake [30,31,32]. Increased fructose ingestion in either humans or animal studies have demonstrated significant hemodynamic changes even after limited periods of time [28,33,34]. Interestingly, the majority of animal studies were unaccompanied by significant increases in body weight, suggesting that factors apart from obesity may contribute to the hypertensive phenotype [30,31]. Chronic animal models using more moderate fructose intake that is consistent with heavy human consumption (15–20% of daily caloric intake) demonstrate cardiovascular and metabolic changes similar to human subjects, although the timeline by which these occur may be skewed [35,36]. The role of endothelial dysfunction has been reviewed in detail [37,38]. Mechanisms involved in the early phases of sodium absorption by the intestine have been studied to a greater extent [39]; however, renal sodium reabsorption [40], the renal renin-angiotensin-aldosterone (RAS) system [41], and sympathetic nervous system [32,42] have received more limited attention. 

## 3. Fructose Influences Sodium Handling and Blood Pressure

### 3.1. Fructose Influences Gastrointestinal Sodium Absorption

Sodium homeostasis is a critical component of blood pressure regulation and has been linked to various cardiovascular and renal complications, including hypertension [43,44,45,46]. Glucose intake is coupled to Na^+^ transport via the luminal sodium-glucose-linked transporter 1 (SGLT1). Intracellular glucose concentration is largely maintained through the glucose transporter 2 (GLUT2) isoform along the basolateral membrane. Chronically, fructose and glucose (but not other sugars) lead to an increase in GLUT2 protein expression along the basolateral membrane [47]. Similar to GLUT5, GLUT2 has a much lower affinity for glucose than other isoforms and therefore functions primarily as a fructose transporter [48]. Fructose transport is facilitated by a downhill concentration gradient between the intestinal lumen and intracellular space [39,49,50]. On the other hand, sodium absorption occurs throughout the small intestine via a variety of transport systems. SLC26A6 (human), also known as the putative anion transporter 1 (PAT1), is a multifunctional apical chloride/base exchanger that increases with fructose feeding in both the jejunum [39,51] and kidney [52]. The function of PAT1 is coupled with that of the intestinal Na/H exchanger 3 (NHE3) so that Na^+^ is reabsorbed with Cl^−^ in an electroneutral manner [53,54]. The presence of fructose amplifies NHE3 function, thereby enhancing absorption of Na^+^ and secretion of H^+^ [39,40,42]. PAT1 also co-localizes with GLUT5 (Slca5), a member of the glucose transporter family, with low affinity for glucose and high affinity for fructose (Figure 1). GLUT5 is the dominant fructose transporter in the jejunum. 

Despite the changes in both sodium and sugar transporters, the impact of fructose on overall gastrointestinal absorption and the resulting fecal excretion of sodium has been given scant investigative attention. An increase in dietary sodium does not increase absolute fecal sodium excretion in Sprague Dawley rats independent of whether the high sodium chow is delivered with glucose or fructose in the drinking water or with water alone. Thus, gastrointestinal sodium absorption increases as dietary intake of sodium increases regardless of the presence or type of sugars in the diet (Figure 1). The task for excretion of the greater total body sodium content relies on the kidney. Notably, urinary sodium excretion is significantly diminished in fructose-fed rats resulting in a positive sodium balance [41]. The renal mechanisms that are understood to date are detailed below.

### 3.2. Fructose Influences Renal Sodium Reabsorption and RAS

The bulk of Na^+^ reabsorption in the mammalian kidney occurs in the proximal tubule, which is responsible for reabsorption of 60–70% of all Na^+^ and fluid filtered by the glomerulus [55]. Similar to the intestine, GLUT2, GLUT5, NHE3, and several isoforms from the SGLT family facilitate the reabsorption of Na^+^. Proximal tubule Na^+^ reabsorption is reliant on secondary active transport by co-transporters such as NHE3 and SGLT isoforms, particularly SGLT2 [56]. Fine tuning of ionic concentrations and gradients is critical to blood pressure homeostasis. Perturbation such as augmented proximal tubule Na^+^ reabsorption results in increased fluid reabsorption, leading to a net positive sodium balance predisposing to hypertension (Figure 1). This mechanism has been linked to hypertension in spontaneously hypertensive rats [57] and Dahl salt-sensitive rats [58]. In carefully executed balance studies, Gordish et al. [41] showed that rats given 20% fructose in their drinking water and placed on high salt diet displayed significantly greater cumulative Na^+^ balance compared with rats given only water or 20% glucose in their drinking water. These findings strongly supported a role for fructose-feeding on renal Na^+^ balance. Notably, except for elevated triglyceride levels in the fructose-fed high salt group, there were no differences in fasting plasma glucose and body weights between fructose-high salt-fed rats and glucose or water controls either with standard or high salt intake [41].

In the proximal tubule, the majority of Na^+^ and all bicarbonate (HCO3^−^) reabsorption occurs via Na^+^/H^+^ exchange, via NHE3 [59]. Acute in vivo microperfusion studies of proximal tubules from Wistar rats exposed to fructose revealed enhanced NHE3 activity. The increase in NHE3 activity was corroborated by in vitro studies showing greater Na^+^ dependent H^+^ flux in the presence of fructose associated with diminished PKA activity in cultured LLC-PK1 cells, a porcine cell line [60]. Fructose, but not glucose, increased NHE3 activity by isolated rat proximal tubules via a PKC-dependent pathway and further potentiated the effects of picomolar concentrations of angiotensin II (Ang II) to increase Na^+^/H^+^ exchange. Na/K-ATPase activity did not change [40]. Moderate amounts of fructose consumption (approximately 40% of daily caloric intake provided as a 20% fructose solution in drinking water) increased tail cuff blood pressure in rats. Proximal tubule expression levels of NHE3 increased significantly in the fructose-fed compared to that of control rats, whereas α1 subunit of the basolateral Na/K-ATPase did not. Moreover, proximal tubules isolated from these rats also showed enhanced Na^+^ reabsorption that was potentiated by Ang II [61,62]. 

The proximal tubule thus appears to become sensitized to Ang II via a PKC mechanism, to which the addition of high Na^+^ intake leads to substantial reabsorption rates. Further, this sensitization extends beyond the proximal tubule into the thick ascending limb and other Na^+^ transport mechanisms in the distal nephron although studies have reported some conflicting results. An early study with 65% dietary fructose failed to observe a change in overall Na^+^/K^+^/2Cl^−^ co-transporter 2 (NKCC2) abundance in the kidney [63]. In vitro and in vivo studies have shown similar effects on NKCC2 expression and activity. Acute in vitro administration of fructose to the thick ascending limb increases NKCC2 activity by increasing protein expression and not phosphorylation along the apical membrane [64]. Chronic in vivo studies using furosemide, an NKCC2 antagonist, to measure acute diuretic and natriuretic responses following NKCC2 blockade found that high fructose-induced hypertension led to significant increases in urine output as well as in urinary potassium, chloride, and sodium concentration. This was reflected by significant increases in NKCC2 mRNA and protein expression [65]. Together, these alterations in transporter expression and activity throughout the nephron facilitate plasma volume expansion which is responsible, at least in part, for the observed hypertension.

The importance of sensitization to Ang II also cannot be understated. Several preclinical studies have shown that even moderate consumption of fructose and Na^+^ can have considerable adverse effects on blood pressure [41,42,61,66]. With the profound increases in Na^+^ retention observed in the various models of fructose-induced hypertension with expected extracellular volume expansion, it would be anticipated that plasma renin activity (PRA) would be suppressed. Further investigation by others reported blunted suppression of PRA in Sprague Dawley rats given 20% fructose for one week, followed by one week of fructose plus 4% NaCl diet [41,66]. In contrast, extending the high salt intake period to three weeks, Soncrant et al. confirmed the elevation in blood pressure using telemetric monitoring and were able to demonstrate inhibition of PRA and plasma Ang II in fructose-fed rats [42]. Thus, it appears that PRA which is typically inhibited by either high blood pressure or expanded extracellular volume requires a longer period of exposure to either of these inhibitory influences in the context of high fructose intake. In other words, a much greater expansion of extracellular volume and, therefore a longer period of positive cumulative Na^+^ balance, may be required to inhibit renin secretion with concomitant fructose-feeding. 

It is also possible that intrarenal RAS contributes to hypertension and increased Na^+^ reabsorption with fructose feeding. Early studies indicated that renal tissue renin expression was suppressed in fructose-fed mice [39] yet increased in rats fed 20% fructose for 12 weeks [65]. Renal tissue Ang II levels were not statistically different from control glucose-fed rats on high salt diet for three weeks despite higher blood pressure [42]. Notably, renal angiotensin AT1 receptor mRNA was increased in adipose tissue but not kidney after three weeks of high fructose ingestion [67]. When fructose was given in greater amounts (60 to 66% in the drinking water) and for longer periods of time (8 to 16 weeks), both Ang I and Ang II as well as AT1 receptor protein expression was increased in kidney tissue [68,69]. Alternatively, the third influence on renin secretion involves sympathetic inputs to the macula densa. Indeed, switching from standard diet (0.4% NaCl) to a high salt (4% NaCl) diet, in fructose-fed rats, increased renal sympathetic nerve activity by ~50% [42]. Denervation of the kidneys bilaterally using cryo-techniques further decreased the already suppressed PRA suggesting sympathetic inputs to the kidney were responsible for the enhanced renin secretion. Notably, tissue Ang II content was not altered by any of the diets compared with controls [42].

In addition to the experimental dietary manipulations, some of the variability in these studies likely stems from methodology used such as the technique for blood pressure assessment (tail cuff vs. telemetry), the time when fructose is initiated relative to the high salt diet, the length of exposure to each component of the diet, and the collection of blood or tissue for measurements of renin, plasma Ang II or their tissue levels. Regardless of the impact of these factors, the findings that a diet enriched for fructose sensitizes the nephron to Ang II such that even minimal activation of the RAS can have profound sodium reabsorption effects.

### 3.3. Fructose Influences the Renal Sympathetic Nervous System

The scientific interest in sympathetic innervation of the kidney has increased recently after the demonstration of marked reductions in blood pressure in individuals with resistant hypertension after denervation using the catheter-based approach [70,71]. Renal sympathetic nerve activity (RSNA) is significantly increased in many forms of experimental hypertension (reviewed by Osborn and Foss [72]), underlying the vital role of RSNA in blood pressure regulation. The mechanisms of increased RSNA in the pathogenesis of hypertension include increased tubular sodium reabsorption and water retention, decreased renal blood flow and glomerular filtration rate, and increased renin secretion from the juxtaglomerular cells which activates RAS [73]. Since moderately high dietary fructose and salt cause hypertension via similar mechanisms, it has been hypothesized that sympathetic activity increases in this dietary milieu. In fact, ingestion of fructose has been shown to result in changes in secretion of hormones that regulate energy balance [74], a shift associated with increased sympathetic nervous system activity [75]. The hormone leptin which promotes satiety is released from adipose tissue cells to maintain energy balance by promoting satiety. In contrast, ghrelin opposes this action and promotes hunger. Ingestion of fructose leads to simultaneously increased secretion of ghrelin and decreased secretion of leptin; however, after chronic ingestion of excess dietary fructose (over one to four weeks), fasting plasma concentrations of leptin are significantly higher [76]. The mechanistic basis for elevated leptin levels likely stems from fructose-induced fatty acid re-esterification and synthesis of VLDL-triglycerides. This pathway causes a shift towards the fat-storing mode, prompting the adipose cells to respond by elevating leptin production. Simultaneously, augmentation of RSNA occurs initiating the onslaught of hypertension-promoting effects. 

Multiple studies have shown that uncontrolled increases in RSNA contribute to both the onset and maintenance of hypertension in both humans and in animal models [72,77], suggesting that coupled with the metabolic changes and energy balance shifts, fructose-induced salt-sensitive hypertension also likely leads to increased RSNA. When rats fed 20% fructose were switched to the high salt diet for one week, they developed hypertension, accompanied by net positive sodium balance, but persistently high PRA [41]. The inability of net positive sodium retention and elevated mean arterial pressure to reduce PRA suggests that increased RSNA is acting upon the juxtaglomerular apparatus to secrete renin and activate RAS. In fact, a preclinical study demonstrated a direct involvement of increased RSNA in fructose-induced salt-sensitive hypertension in conscious rats chronically instrumented with nerve telemetry devices [42]. High salt intake in rats on 20% fructose, but not glucose, resulted in a 41% increase in RSNA after 1 week. Further confirmation of the role of RSNA was provided by demonstrating a decrease in blood pressure after bilateral renal denervation using the cryo-technique. Although the rats did not display the full complement of metabolic syndrome, insulin sensitivity was also reduced in the hypertensive fructose-fed rats. This restoration of RSNA to baseline following cryoablation was accompanied by decreased blood pressure and improved insulin sensitivity [42]. 

The attenuation of PRA suppression in fructose and high salt-fed rats is mitigated when the rats ingest a high salt diet for at least three weeks. Therefore, it appears that the initial mechanism of fructose-induced salt-sensitive hypertension involves persistently elevated PRA that eventually subsides after two to three weeks of high salt feeding. During this period augmentation of RSNA is initiated, which serves as a feed-forward mechanism that takes over when PRA returns to baseline in order to maintain elevated arterial pressure (Figure 1). 

Concurrently, an indirect mechanism involving reactive oxygen species generation may be partially responsible for blunted suppression of PRA initially, followed by increases in RSNA. High fructose and salt feeding in rats leads to increased oxidative stress as measured by 8-isoprostane excretion, and supplementation with the superoxide dismutase mimetic, Tempol, attenuates both the increase in mean arterial pressure and reactive oxygen species generation [66]. Furthermore, oxygen radicals can activate the sympathetic sites that are involved in the pathogenesis of hypertension [78]. Thus, moderately high fructose and salt feeding in rats causes hypertension in at least two phases. Firstly, an increase in blood pressure results from the direct action of renin, which eventually returns to baseline as a result of the negative feedback mechanism evoked by the net positive, albeit greater, cumulative sodium balance and increase in blood pressure. Additionally, generation of reactive oxygen species contributes to high PRA. Secondly, after two to three weeks of high salt feeding, PRA decreases and hypertension is maintained via increased RSNA. Oxygen radicals further contribute to sympathoexcitation, providing a collateral mechanism to maintain increased blood pressure.

## 4. Conclusions

High fructose and salt intakes are contributing to the increased incidence and prevalence of hypertension in the U.S. and globally. Even prior to the development of full blown metabolic syndrome, fructose-induced enhancement of renal Na^+^ reabsorption with greater positive net cumulative Na^+^ balance can lead to elevations of blood pressure. The inhibition of PRA is also blunted despite higher blood pressure and extracellular volume and is driven by enhanced renal sympathetic nerve inputs. Activation of circulating RAS, therefore, further augments blood pressure. After longer periods of fructose and high salt ingestion, intrarenal RAS is also increased which may further drive tubular Na^+^ reabsorption. The combined inputs from afferent nerves, reactive oxygen species and increased metabolic hormones such as leptin work centrally to stimulate sympathetic outputs and further increase blood pressure (Figure 1). Thus, combined fructose and high salt diets may lead to hypertension and increased cardiovascular risks even short term and in individuals without the full metabolic syndrome.

## Figures and Tables

**Figure 1 nutrients-11-00569-f001:**
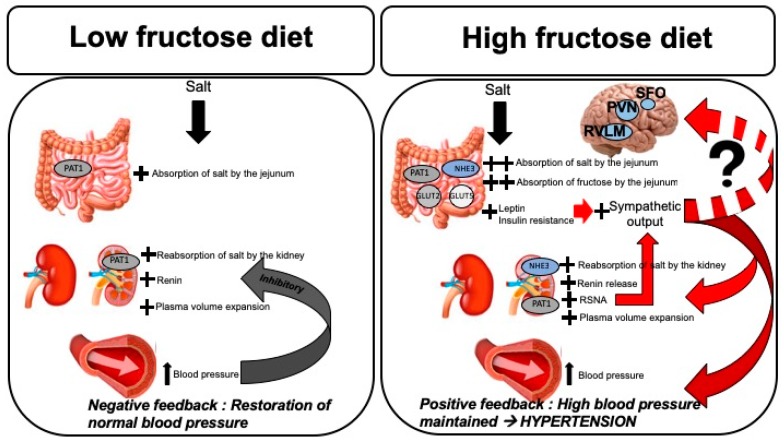
Mechanisms of fructose-induced salt-sensitive hypertension. Left panel: In the absence of high dietary fructose, absorption of salt in the small intestine (jejunum) and proximal tubule of the kidney is accomplished by PAT1. Increased sodium load leads to plasma volume expansion which activates RAS, resulting in elevation of blood pressure. Either increased plasma renin activity or increased blood pressure alone are capable of initiating the negative feedback mechanism to dampen NaCl reabsorption and decrease RAS activation, resulting in restoration of blood pressure back to normal. Right panel: When dietary fructose intake is high in conjunction with high salt, intestinal fructose absorption with Glut 5 and Glut 2 and sodium is absorption via PAT1 and NHE3 in increase. Proximal tubular sodium reabsorption is increased by both PAT1 and NHE3. Elevated fructose load leads to increased levels of circulating leptin and insulin, resulting in insulin resistance. These hormones lead to increased sympathetic outputs. In addition to activation of RAS and Na^+^ reabsorption, increased RSNA also participates in elevating blood pressure, ultimately resulting in the loss of the negative feedback mechanism. Increased afferent inputs to the central barosensitive regions may also contribute to hypertension by creating a feedforward situation via efferent sympathetic nerves. PAT1, putative anion transporter 1; NHE3, sodium/hydrogen exchanger 3; Glut 5 and Glut 2, glucose transporters 5 and 2, respectively; RAS, renin-angiotensin-aldosterone system; RSNA, renal sympathetic nerve activity; SFO, subfornical organ; PVN, paraventricular nucleus; RVLM, rostral ventrolateral medulla.

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
