# Peer review of "Hypertension Associated with Fructose and High Salt: Renal and Sympathetic Mechanisms"

_nutrients, 2019, doi:10.3390/nu11030569_

Reviewer 1 Report

I was honored to review the manuscript entitled " Hypertension Associated with Fructose and High Salt: Renal and Sympathetic Mechanisms" submitted to Nutrients.

 Taking into account the multiple studies ongoing in the field of hypertension this type of review is needed.  I have only few small remarks that authors should adress properly.

  Points that need correction:

  - please provide the list of abbreviations.

  - please provide the flow diagram of the data search and the methodology of data search with the quantity of papers searched.

  - Introduction and Discussion section needs improvement- please cite: doi: 10.3390/nu10091284. ; doi: 10.26402/jpp.2018.2.13. ; Int J Clin Exp Med. 2015; 8(7): 10358–10366. ; Biomed Environ Sci. 2016 Oct;29(10):706-712.

  - It would be also useful to illustrate some of the explained mechanisms.

 - In general the manuscript is quite long and therefore a little bit difficult to follow. Thus, please provide some data in form of tables.

 Author Response

Reviewer #1

Thank you for your kind words regarding this review of fructose, the sympathetic nervous system and blood pressure. We appreciate your comments toward refining our manuscript.

1.       Provide the list of abbreviations

Response: We have provided the list of abbreviations at the end of the article

 2.       Provide the flow diagram of the data search and the methodology of data search with the quantity of papers searched.

Response: Since this is a review rather than a meta-analysis, we have not included a flow diagram for our search. We have provided a section detailing our search parameters and the papers that were considered.

 3.       Introduction and Discussion needs improvement. Please cite….

Response: We have modified the introduction and section on fructose consumption (Section 2) with greater emphasis on human studies. Hopefully this puts the more basic preclinical studies in perspective.

 4.       Useful to illustrate some of the explained mechanisms.

Response: The mechanism is illustrated in Figure 1 and detailed in the legend.

 5.       In general the manuscript is quite long…please provide some data in form of tables.

Response: We have tried to place some data in tables but the preclinical data did not lend itself to this format.

Reviewer 2 Report

The manuscript "Hypertension associated with fructose and high salt: renal and sympathetic mechanisms" presents very interesting review on the relationship between fructose intake and hypertension and its underlined mechanisms. I have annotated the manuscript with several corrections, which I believe will improve the readability of the paper. 

Since Nutrients focus on human nutrition, evidence from human studies should also reviewed. For example, systematic review and meta-analysis regarding fructose consumption and hypertension has been published in 2014 (J Am Coll Nutr. 2014 Jul 4; 33(4): 328–339). According to this review, total fructose intake was not associated with an increased risk of hypertension in 3 large prospective cohorts of U.S. men and women. Is this result different for example according to the countries? How important is fructose compared with other risk factors? Please discuss the newest evidence in human.

In addition to that, with high fructose diet, about how much of salt absorption is increased? It may help readers to understand this mechanisms and its impact on hypertension.

Author Response

Thank you for your thoughtful review of our manuscript. We have revised the manuscript accordingly.

 1.       I have annotated the manuscript…improve the readability of the paper.

Response: Unfortunately, we did not receive the copy of the paper that you so kindly annotated and edited. I hope our edits have made it easier to read.

 2.       Since Nutrients focuses on human nutrition evidence from human studies should also be reviewed. …meta-analysis. Total fructose intake was not associated with increased risk of hypertension…men and women.

Response: We now cite this study. Importantly, a study by the same authors reported a 12% increase in risk of hypertension when they included the CARDIA cohort which includes young individuals over a very long observational period. We cite this study as well.

 3.       How important is fructose compared with other risk factors?

Response: The relative risk of hypertension associated with fructose intake is now placed into perspective of other risk factors. This can be found in Section 2.

 4.       In addition…with high fructose diet about how much of salt absorption is increased?

Response: We have added as a discussion of this in Section 3.2. Balance studies in humans are not available. In one carefully done rat study, an increase in dietary sodium did not increase sodium excretion fecal sodium excretion when delivered with either glucose or fructose.  In fact, after 2 weeks of control _ high salt or sugars + high salt diet, the amount of sodium excreted did not differ from baseline. However, the percent of fecal sodium excretion declined from 3.6 – 4.0 to 0.4-0.5% consistent with an increase in gastrointestinal sodium absorption in the presence of high salt. There was no apparent effect of concurrent intake of either glucose or fructose. Overall this suggested that handling of the enhanced absorption of salt requires renal excretion regardless of whether salt is ingested with water or with either glucose or fructose. However, renal excretion of salt is diminished when fructose is ingested (Gordish, et al, 2018; Ref. 33 in original manuscript).

Round  2

Reviewer 2 Report

Thank you for revising the manuscript. The manuscript has been revised carefully according to the comments.